**Data Availability Statement:** All relevant data are within the manuscript and its Supporting information files.

# The incidence, characteristics, impact and risk factors of post-COVID chronic pain in Thailand: A single-center cross-sectional study

Suratsawadee Wangnamthip[1], Nantthasorn Zinboonyahgoon[1]*, Pranee Rushatamukayanunt[1], Patcha Papaisarn[1], Burapa Pajina[1], Thanawut Jitsinthunun[1], Panuwat Promsin[2], Rujipas Sirijatuphat[2], César Fernández-de-las-Peñas[3,4], Lars Arendt-Nielsen[3,5,6], Daniel Ciampi de Andrade[3]

1 Faculty of Medicine Siriraj Hospital, Department of Anesthesiology, Mahidol University, Bangkok, Thailand,
2 Faculty of Medicine Siriraj Hospital, Department of Medicine, Mahidol University, Bangkok, Thailand,
3 Faculty of Medicine, Department of Health Science and Technology, Center for Neuroplasticity and Pain (CNAP), Aalborg University, Aalborg, Denmark, 4 Department of Physical Therapy, Occupational Therapy, Rehabilitation and Physical Medicine, Universidad Rey Juan Carlos, Alcorcón, Spain, 5 Department of Gastroenterology & Hepatology, Mech-Sense, Clinical Institute, Aalborg University Hospital, Aalborg, Denmark, 6 Steno Diabetes Center North Denmark, Clinical Institute, Aalborg University Hospital, Aalborg, Denmark

* nantthasorn@gmail.com

## Abstract

The COVID-19 pandemic has affected millions of individuals worldwide. Pain has emerged as a significant post-COVID-19 symptom. This study investigated the incidence, characteristics, and risk factors of post-COVID chronic pain (PCCP) in Thailand. A cross-sectional study was conducted in participants who had been infected, including those hospitalized and monitored at home by SARS-CoV-2 from August to September 2021. Data were collected for screening from medical records, and phone interviews were done between 3 to 6 months post-infection. Participants were classified into 1) no-pain, 2) PCCP, 3) chronic pain that has been aggravated by COVID-19, or 4) chronic pain that has not been aggravated by COVID-19. Pain interference and quality of life were evaluated with the Brief Pain Inventory and EuroQol Five Dimensions Five Levels Questionnaire. From 1,019 participants, 90% of the participants had mild infection, assessed by WHO progression scale. The overall incidence of PCCP was 3.2% (95% CI 2.3–4.5), with 2.8% (95% CI 2.0–4.1) in mild infection, 5.2% (95% CI 1.2–14.1) in moderate infection and 8.5% (95% CI 3.4–19.9) in severe infection. Most participants (83.3%) reported pain in the back and lower extremities and were classified as musculoskeletal pain and headache (8.3%). Risk factors associated with PCCP, included female sex (relative risk [RR] 2.2, 95% CI 1.0–4.9) and greater COVID-19 severity (RR 3.5, 95% CI 1.1–11.7). Participants with COVID-19-related exacerbated chronic pain displayed higher pain interferences and lower utility scores than other groups. In conclusion, this study highlights the incidence, features, and risk factors of post-COVID chronic pain (PCCP) in Thailand. It emphasizes the need to monitor and address PCCP, especially in severe cases, among females, and individuals with a history of chronic pain to improve their quality of life in the context of the ongoing COVID-19 pandemic.

**Funding:** This project was supported by the Siriraj Research and Development Fund, Faculty of Medicine Siriraj Hospital, Mahidol University (grant number: R016531012; https://www2.si.mahidol.ac.th/en/research/fund_1/). SW, NZ, PR, TJ, PP, and RS received support from a Chalermphrakiat Grant from the Faculty of Medicine Siriraj Hospital, Mahidol University. The Center for Neuroplasticity and Pain was supported by the Danish National Research Foundation, Copenhagen, Denmark (grant number: DNRF121; https://dg.dk/en/). DCA received support from a Novo Nordisk Grant, Hellerup, Denmark (grant number: NNF21OC0072828; https://novonordiskfonden.dk/en/grant/). LA and CF received support from a Novo Nordisk Grant, Hellerup, Denmark (grant number: NNF21OC0067235; https://novonordiskfonden.dk/en/grant/) for the study "Incidence and Characterisation of Persistent Pain in COVID-19 Survivors: A pan-European Concerted Action." The funders had no role in study design, data collection and analysis, decision to publish, or preparation of the manuscript.

**Competing interests:** The authors have declared that no competing interests exist.

## Introduction

The severe acute respiratory syndrome coronavirus 2 (SARS-CoV-2) initiated the pandemic of the coronavirus disease 2019 (COVID-19), and its long-term consequences are still under investigation. The COVID-19 pandemic has had significant impacts on various aspects of human life, especially among those directly affected. These effects extend beyond physical health, encompassing a range of psychological outcomes [1,2]. Furthermore, implementing lockdown measures and isolation has brought about additional adverse consequences for overall health [3].

Many studies have shown the long-term symptoms of SARS-CoV-2 infection, referred to as post-COVID-19 condition or long-COVID. Long-term symptoms include persistent fatigue, dyspnea, anosmia, ageusia, and also chronic pain (musculoskeletal pain and headache), which have now persisted as long as two-three years after infection [4]. As pain is a notable consequence of COVID-19 [1,5]. Soares et al. provided the first controlled assessment of prevalence and characteristics of post-COVID pain in COVID-19 participants in Brazil [6]. They found that the incidence of de novo pain after SARS-CoV-2 infection was as high as 65.2%, with de novo headache affecting 39.1% and new-onset chronic pain at 19.6% of patients. They also reported post-COVID pain to be more frequent in the head/neck and lower extremities [6].

The exact mechanisms of post-COVID pain are still unknown and might develop from multiple factors. It has been hypothesized that peripheral and central sensitization plays an important role in post-COVID-related pain [7,8]. Moreover, other factors, including psychological, environmental, associated comorbidities, and impaired muscle metabolism, are also involved [8] A cluster analysis study of post-COVID found that higher symptom severity at the COVID-19 acute phase and a greater number of pre-existing comorbidities were associated with a greater likelihood of post-COVID symptomatology [9]. A meta-analysis found that the prevalence of musculoskeletal post-COVID pain was 17.3% (95% CI 11.1–25.8) and reported that the pooled prevalence of post-COVID symptoms was higher in Asia than in Europe and other geographical regions [10]. Similar prevalence rate was reported by a meta-analysis focusing just on the prevalence of post-COVID pain [11]. As the incidences of post-COVID pain have been estimated at different time points across studies the possible recovery over time is a factor to consider when comparing studies. Furthermore, the incidence also to a large degree depends on whether the patients have recovered at home or have been hospitalized [11].

Given the limited available data on post-COVID persistent symptoms in Thailand, the aims of the current study were 1) to investigate the incidence of chronic post-COVID pain after a SARS-CoV-2 acute infection, including subjects hospitalized and monitored at home, 2) to record pain characteristics and its impact of individuals with post-COVID pain, and 3) to identify the risk factors associated with the development of post-COVID chronic pain in a population in Thailand.

## Materials and methods

### Study design and participants

This was a single-center cross-sectional study including COVID-19 patients who have tested positive for SARS-CoV-2 from August 2021 to September 2021 and were admitted to the Siriraj hospital (a tertiary care center), secondary hospitals (hospitals for patients with less severe symptoms) or managed at home isolation program (hospital-monitored). Exclusion criteria were patients who could not understand Thai, refused to participate, and could not complete information during the interview.

This study was conducted in accordance with the International Council on Harmonization's Good Clinical Practice, Declaration of Helsinki, and Belmont Report. With Siriraj Institutional Review Board approval (Si 873/2021), the research team identified patients who met the study criteria from the hospital database; then, patients were informed about study protocol and asked to consent to participate. Considering the study's minimal risk and its reliance on phone interviews as the primary mode of data collection, the SIRB approved the use of verbal informed consent. Phone interviews were performed by healthcare providers using standardized conversations and questionnaires (S2 Appendix).

## Phone interview protocol

Participants were assessed through telephone interviews by standardized trained interviewers who were equipped with informative guidance and practical exercises for structured interviews. Subsequently, pain specialists assessed interrater reliability, with each session lasting a minimum of one day of training to reduce non-response bias [12,13]. The participants were interviewed between 3 and 6 months after the confirmed positive RT-PCR test. The structured interview script encompassed identifying research staff, introducing study aims, and adhering to the script throughout the interview. A routine callback in case of failed contact protocol was implemented [14]. Individual telephone number was contacted up to 6 times on different weekdays and different periods between 8 am and 9 pm (Bangkok time). The telephone numbers were randomly chosen from the hospital database using a random function in Excel program and conducted from November 2021 to March 2022. The refusal rate and causes of not participating were recorded.

## Data collection

**Demographic and COVID-19 clinical data.** According to the review of chronic pain-associated factors, the following demographic variables, including age, gender, body weight, height, race, level of education, and occupation, medical comorbidities, mental health status (major depressive disorder, generalized anxiety disorder), and history of chronic pain (persistent pain for longer than 3 months before the infection) were interviewed and collected [15]. The clinical information associated with the COVID-19 acute phase, including information related to a hospital stay, severity of disease, and COVID-19 vaccine status, was obtained from the medical records. Patients were categorized into three groups according to the severity of COVID-19 disease from WHO Clinical Progression Scale. "Mild" was described as having mild symptoms and not requiring oxygen supplementation. "Moderate" was described as moderate symptoms which require an oxygen cannula to maintain SpO2 > 94%. "Severe" was defined as severe symptoms and need to use of high flow nasal cannula, non-invasive ventilator or mechanical ventilator to support the symptoms or required vasopressor or inotropic drugs [16]. All data were collected by a secured computer system that could be accessed only by the research team.

**Post-COVID pain symptomatology.** Symptoms experienced after the acute infection were interviewed at the time of assessment, including pain, fatigue, and anosmia. Pain was defined as physical suffering or discomfort experienced by the patient. Fatigue was defined as the feeling of being severely overtired or lack of energy. Anosmia was defined as the total loss of sense of smell. Participants answered yes/no to the presence of these symptoms. The presence of fatigue and anosmia was assessed because of their high comorbidity with post-COVID pain [11]. In addition, participants with pain were further interviewed to identify whether it was a de novo pain symptom after COVID-19 or chronic pain aggravated after COVID-19.

Participants were classified into four groups according to post-COVID pain: Group 1 (no pain) was defined as no post-COVID pain and no history of chronic pain before the infection; Group 2 (PCCP) was defined as participants experiencing de novo post-COVID pain symptomatology for more than 3 months after the infection; Group 3 (chronic pain that has been aggravated by COVID-19) was defined as participants with a history of chronic pain before the infection and this pain was aggravated after; and Group 4 (chronic pain that has not been aggravated by COVID-19) defined as participants with a history of chronic pain, this pain was not aggravated by COVID-19 and no further post-COVID pain was developed. Only participants in Groups 2, 3, and 4 were interviewed about their pain characteristics, location, and type. Then researchers categorized it as chronic pain according to the International Classification of Diseases 11th Revision (ICD-11) [17].

**Quality of life.** The EuroQol Five Dimensions Five Levels Questionnaire (EQ-5D-5L) was applied to assess the quality of life of all participants. To measure the level of problems in each quality-of-life, five categories (mobility, self-care, usual activities, pain/discomfort, and anxiety/depression) using a 5-Likert scale (no problems, slight problem, moderate problems, severe problems, unable to/extreme situation) were assessed and used to calculate the utility score. The EQ-5D-5L utility score ranges from 0 (representing death) to 1 (representing total health). Some individuals can rate their health as "worse than death," making a score less than 0 possible. The range of utility score of Thai population is between -0.4 to 0.9 [18]. Participants were also asked to report their overall health using a 100 mm visual analog scale (VAS) called "EQ-5D-VAS" with "worst imaginable health" and "best imaginable health" as the endpoints [19]. The EQ-5D-VAS is conceptually different from the EQ-5D utility index since it represents the overall patient perspective on quality of life.

**Pain interference.** The Thai version of the Brief Pain Inventory (BPI) was used to determine pain interference or pain impact within the last 24 hours regarding general activity, mood, walking ability, normal work, relations with other people, sleep, and enjoyment of life in those participants who had experienced new pain after COVID-19 (Group 2), had a history of chronic pain being aggravated by COVID-19 (Group 3), or had a history of chronic pain without being aggravated by COVID-19 (Group 4). Each category ranges from 0–10, where 0 indicates no interference and 10, maximum interference. The total BPI score ranges "from 0–70" [20].

## Sample size calculation

Soares et al. reported that post-COVID pain incidence among hospitalized patients was 19.6% [6]. We expected an incidence of PCCP to be 20% (0.2), 95% confidence interval (CI), and 15% (d = 0.03) error. Based on these data, the estimated sample size was 673. We estimated a dropout rate of 30%; therefore, the total sample size was estimated to 1,000.

## Data analysis

Analyses were performed using SPSS® statistical package 18.0 (SPSS®, Inc., Chicago, IL) and Stata version 14.1 (StataCorp [2015]. Stata Statistical Software: Release 14. College Station, TX: StataCorp LP). The continuous data are reported as mean ± standard deviation (SD) and categorical data are reported as numbers and percentages in demographic data. The normality was assessed using Shapiro Wilk test with a p-value of 0.05. The incidence of de novo PCCP, fatigue, and anosmia in a Thai population was reported as a number and percentage with a 95% CI.

Demographic data between four groups (Groups 1–4) were compared using one-way analysis of variance (ANOVA) for continuous data and Chi-Square or Fisher's Exact test for

categorical data. A p-value less than 0.05 demonstrates statistical significance. The risk of developing PCCP (Group 2), and chronic pain that has been aggravated by COVID-19 (Group 3) are reported with relative risk (RR) with 95% CI in relation to sex, age, vaccination status, psychological and physical comorbidities, and presence of pre-infection chronic pain. The pain distribution, including pain location and diagnosis according to ICD-11 were reported as number and percentages. Those significant variables were included in generalized linear models to adjust RR in finding the significant risk factors of developing PCCP and chronic pain aggravated by COVID-19. The positive predictors are defined with a p-value < 0.05 and presented as RR with a 95% CI. Missing data were managed through listwise deletion, where records with missing data for any variable of interest were excluded from the analysis.

# Results

## Participants

From 1,490 participants selected from hospital/home monitoring database with confirmation of SARS-CoV-2 infection (RT-PCR) from August to September 2021 and started phone interviewing between November 2021 and March 2022, a total of 1,019 individuals were finally analyzed (S1 Appendix). The 471 individuals were dropped out due to inability to contact, death, refusal to participate, and inability to understand the Thai language properly (Fig 1). The demographic data are shown within Table 1. Most participants (93.7%) were Thai people and the remaining were from Laos (2.9%), Myanmar (2.5%), ethnic groups (0.7%) and Cambodian (0.2%). In addition, 33.4% had not been vaccinated before being infected. Most participants were classified as mild COVID-19 severity (89.7%) (Table 2), and the duration from SARS-CoV-2 infection to interview ranged from 94 to 184 days (mean ± SD: 137.3 ± 20.9 days).

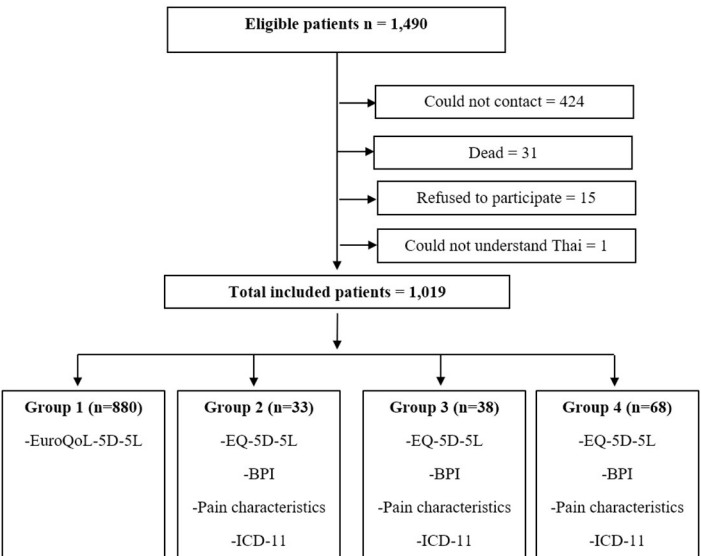

**Fig 1. Recruitment and data collection diagram.** Abbreviations: EQ-5D-5L, EuroQol 5 Dimensions 5 Levels; BPI, Brief Pain Inventory; ICD-11, International Classification of Diseases 11th Revision.

**Table 1. Demographic data of the total sample (n = 1,019).**

| | |
|---|---|
| Age (years), mean ± SD | 46.5 ± 17.2 |
| Male, n (%) | 419 (41.1) |
| BMI (kg/m$^2$), mean ± SD | 25.4 ± 6.5 |
| Thai race, n (%) | 955 (93.7) |
| Marital status, n (%) | |
| Married | 564 (55.3) |
| Widow/divorce | 110 (10.8) |
| Single | 345 (33.9) |
| Graduated high school and below, n (%) | 751 (73.7) |
| Pre-existing comorbidities, n (%) | 457 (44.8) |

Abbreviations: BMI, Body Mass Index; SD, Standard Deviation.

## Incidence of post-COVID chronic pain

Overall, 86.4% of participants did not report any post-COVID pain (Group 1). The incidence of PCCP (Group 2) was 3.2% (95% CI 2.3–4.5). Nevertheless, the incidence of PCCP according to COVID-19 severity was 2.8% (95% CI 2.0–4.2) in those with mild disease, 5.2% (95% CI 1.2–14.2) in those with moderate and 8.5% (95% CI 3.4–19.9) in those who had severe illness. Additionally, the incidence of chronic pain aggravated by COVID-19 (Group 3) was 3.7% (95% CI 2.7–5.1), whereas the prevalence of chronic pain not aggravated by COVID-19 (Group 4) was 6.7% (95% CI 5.3–8.4). The incidence of other post-COVID symptoms was 21.6% (95% CI 19.2–24.2) for fatigue and 7.1% (95% CI 5.7–8.8) for anosmia.

**Table 2. Clinical data.**

| | |
|---|---|
| Severity of infection, n (%) | **(n = 1,019)** |
| Mild | 914 (89.7) |
| Moderate | 58 (5.7) |
| Severe | 47 (4.6) |
| Hospital Admission, n (%) | 661 (64.9) |
| Vaccinated before infection, n (%) | 679 (66.6) |
| History of chronic pain, n (%) | 108 (10.6) |
| Fatigue, n (%) | 220 (21.6) |
| Anosmia, n (%) | 72 (7.1) |
| Group, n (%) | |
| 1. No post-COVID pain and no Hx chronic pain (No pain) | (86.4) |
| 2. De novo Post-COVID Chronic Pain (PCCP) | 33 (3.2) |
| 2.1 Mild COVID-19 severity (n = 914) | 26 (2.8) |
| 2.2 Moderate COVID-19 severity (n = 58) | 3 (5.2) |
| 2.3 Severe COVID-19 severity (n = 47) | 4 (8.5) |
| 3. History of chronic pain and aggravated by COVID-19 (CP with aggravated pain) | 38 (3.7) |
| 4. History of chronic pain and not aggravated by COVID-19 (CP without aggravated pain) | 68 (6.7) |
| EQ-5D-5L, utility score (0–1), mean ± SD | 0.9 ± 0.1 |
| EQ-5D-VAS score (0–100), mean ± SD | 87.6 ± 12.8 |

Abbreviations: EQ-5D-5L, EuroQol 5 Dimensions 5 Levels; VAS, Visual Analog Scale; SD, Standard Deviation.

**Table 3. Comparison between Group 1 (no pain), Group 2 (post-COVID chronic pain, PCCP), Group 3 (chronic pain with aggravated pain), and Group 4 (chronic pain without aggravated pain).**

| Total sample (n = 1,019) | Group 1 (n = 880) | Group 2 (n = 33) | Group 3 (n = 38) | Group 4 (n = 68) | p-value |
|---|---|---|---|---|---|
| Age, mean ± SD | 45.2 ± 16.9 | 49.2 ± 14.3 | 51.5 ± 16.7 | 59.1 ± 16.4 | < 0.001[a] |
| BMI (kg/m$^2$), mean ± SD | 25.2 ± 6.5 | 28.1 ± 8.9 | 24.9 ± 5.8 | 26.8 ± 5.4 | 0.021[a] |
| Female, n (%) | 503 (57.2) | 25 (75.8) | 27 (71.1) | 46 (67.6) | 0.024[a] |
| Marital status, n (%) | | | | | < 0.001[a] |
|   Married | 482 (54.8) | 20 (60.6) | 21 (55.3) | 41 (60.3) | |
|   Divorced / Widow | 80 (9.1) | 4 (12.1) | 5 (13.2) | 21 (30.9) | |
|   Single | 318 (36.1) | 9 (27.3) | 12 (31.6) | 6 (8.8) | |
| Graduated high-school and below, n (%) | 648 (73.6) | 27 (81.8) | 27 (71.1) | 49 (72.1) | 0.716 |
| Pre-existing comorbidities, n (%) | 366 (41.6) | 21 (63.6) | 21 (55.3) | 50 (73.5) | < 0.001[a] |
| Mental health problems, n (%) | 17 (1.9) | 1 (3.0) | 8 (21.1) | 5 (7.4) | < 0.001[a] |
| History of chronic pain, n (%) | 0 (0.0) | 4 (12.1) | 38 (100) | 68 (100) | < 0.001[a] |
| Severity of COVID-19 infection, n (%) | | | | | < 0.001[a] |
|   Mild (n = 914) | 813 (92.4) | 26 (78.8) | 26 (68.4) | 49 (72.1) | |
|   Moderate (n = 58) | 36 (4.1) | 3 (9.1) | 10 (26.3) | 9 (13.2) | |
|   Severe (n = 47) | 31 (3.5) | 4 (12.1) | 2 (5.3) | 10 (14.7) | |
| Hospital admission, n (%) | 558 (63.4) | 24 (72.7) | 25 (65.8) | 54 (79.4) | 0.045[a] |
| Duration from infection to interview (days), mean ± SD | 137.1 ± 20.9 | 139.1 ± 21.2 | 139.8 ± 23.9 | 138.0 ± 18.8 | 0.801 |
| Length of stay (days), mean ± SD | 5.4 ± 5.6 | 6.6 ± 4.6 | 6.7 ± 5.9 | 7.7 ± 5.8 | 0.005[a] |
| Vaccinated before infection, n (%) | 601 (68.3) | 24 (72.3) | 19 (50.0) | 35 (51.5) | 0.004[a] |
| Fatigue, n (%) | 165 (18.8) | 12 (36.4) | 20 (52.6) | 23 (33.8) | < 0.001[a] |
| Anosmia, n (%) | 63 (7.2) | 3 (9.1) | 4 (10.5) | 2 (2.9) | 0.445 |
| EQ-5D-5L, utility score (0–1) mean ± SD | 0.97 ± 0.1 | 0.87 ± 0.17 | 0.78 ± 0.25 | 0.82 ± 0.21 | < 0.001[a] |
| EQ-5D-VAS score (0–100), mean ± SD | 89.5 ± 11.3 | 77.6 ± 15.5 | 72.3 ± 18.2 | 76.3 ± 13.3 | < 0.001[a] |

Abbreviation: BMI, Body Mass Index; EQ, EuroQol; VAS, Visual Analog Scale; PCCP, Post-COVID Chronic Pain; CP, Chronic Pain; SD, Standard Deviation.
[a]Statistically significant difference (Chi-square test).

Table 3 compares participants' characteristics by group. There were significant differences in age, BMI, gender, marital status, medical comorbidities, baseline mental health problems, history of chronic pain, severity of COVID-19 disease, admission to the hospital, length of stay at hospital, and type of vaccine, fatigue and quality of life (utility and overall score) among the four groups. The post hoc analysis revealed that Group 4 was older than Group 1 (p < 0.001) and Group 2 (p = 0.034). The length of stay at the hospital in Group 4 was considerably longer than in Group 1 (p = 0.007). Group 1 had significantly higher EQ-5D-5L utility and EQ-5D-VAS scores than all pain groups (p < 0.001), whereas Group 2 had significantly higher (p = 0.026) EQ-5D-5L utility and overall scores than Group 3. Individuals within Group 1 had a significantly lower number of moderate and severe infections when compared with all pain groups (p < 0.05). Patients in Groups 3 and 4 were more significantly associated with fatigue than Group 1 (p < 0.05). The presence of anosmia was similar in all groups (Table 3). The vaccine type and its analyses are in S3 Appendix.

## Pain characteristics, ICD-11, and pain impact

The pain-related disability using BPI and ICD-11 diagnoses are shown in Table 4 and S3 Appendix. According to ICD-11 diagnoses, MG30.30 (musculoskeletal pain) was the most

**Table 4. Pain interferences and pain diagnosis according to the ICD-11 in all pain groups.**

| BPI (n = 139) | Group 2 (n = 33) | Group 3 (n = 38) | Group 4 (n = 68) | P value |
|---|---|---|---|---|
| Minimum pain | 2.1 ± 1.5 | 2.4 ± 2.1 | 1.4 ± 1.8 | 0.017[a] |
| Average pain | 3.9 ± 2.1 | 4.8 ± 1.7 | 3.7 ± 2.5 | 0.030[a] |
| Maximum pain | 5.9 ± 2.2 | 7.1 ± 2.1 | 5.5 ± 2.6 | 0.006[a] |
| Current pain | 2.5 ± 2.2 | 3.4 ± 2.7 | 2.4 ± 2.4 | 0.129 |
| Pain interferences; | | | | |
| Total BPI score | 13.9 ± 12.8 | 18.7 ± 17.5 | 13.4 ± 13.8 | 0.266 |
| ICD-11[b] | Group 2 (n = 33) | Group 3 (n = 38) | Group 4 (n = 68) | P-value |
| More than 1 pain diagnosis | 3 (9.1) | 4 (10.5) | 7 (10.3) | 0.977 |
| | (n = 36) | (n = 42) | (n = 75) | |
| MG30.03 Primary headache | 3 (8.3) | 6 (14.3) | 5 (6.7) | 0.383 |
| MG30.2 Chronic post-surgical pain syndrome | 0 (0) | 1 (2.4) | 0 (0) | 0.264 |
| MG30.3 Musculoskeletal pain | 30 (83.3) | 32 (66.7) | 58 (77.3) | 0.708 |
| MG30.4 Visceral pain | 1 (2.8) | 0 (0) | 1 (1.3) | 0.560 |
| MG30.51 Neuropathic pain | 2 (5.6) | 2 (4.8) | 11 (14.7) | 0.139 |
| MG30.6 Orofacial pain | 0 (0) | 1 (2.4) | 0 (0) | 0.264 |

Abbreviations: ICD-11, International Classification of Diseases 11th Revision; BPI, Brief Pain Inventory.

[a] Statistically significant difference (Chi-square test).

[b] Each pain diagnosis of the multiple has been included on each group.

frequent in all groups. After evaluating the PCCP group, 30 (83.3%) participants complained of generalized muscle pain (MG30.3), 3 (8.3%) had primary headache (MG30.03), 2 (5.6%) complained of symptoms of diabetic peripheral neuropathy at both feet (MG30.51), and one patient reported epigastric pain due to gastritis (MG30.4Z). No differences in ICD-11 diagnoses among the pain groups were observed (Table 4). Regarding the location of pain, back and leg pain was the area most frequently affected in all groups (40%-68%), although the presence of pain in more than one area (generalized pain pattern) was also frequent.

According to pain interferences, minimum, average, and maximum pain show between-groups differences. Post hoc analyses showed that participants in Group 3 reported higher minimum pain intensity (2.4 ± 2.1 vs 1.4 ± 1.8, p = 0.021), average pain intensity (4.8 ± 1.7 vs 3.7 ± 2.5, p = 0.027), and maximum pain intensity (7.1 ± 2.1 vs 5.5 ± 2.6, p = 0.006) scores than participants in Group 4. The pain interferences from total BPI tend to be highest in Group 3 but did not reach statistical significance (Table 4).

### Factor associated with de novo post-COVID chronic pain

Table 5 compares demographic data between no pain (Group 1) and PCCP (Group 2) and shows the relative risk of suffering from de novo PCCP. The univariate analysis revealed that female sex (RR 1.3, 95% CI 1.1–1.6, p = 0.006), having previous medical diseases (RR 1.5, 95% CI 1.2–2.0, p = 0.002), and greater severity of the COVID-19 condition (RR 3.63, 95% CI 1.4–9.6, p < 0.009) were significantly associated with the development of PCCP. After adjusting relative risk with generalized linear models, it was shown that female sex (RR 2.2, 95% CI 1.0–4.9, p = 0.044) and severe illness of COVID-19 (RR 3.5, 95% CI 1.1–11.7, p = 0.040) were those risk factors significantly associated with development of PCCP. There is no significant association between admission status, length of stay or vaccination status and the development of PCCP.

**Table 5. Relative risk of suffering from post-COVID chronic pain (PCCP, Group 2) in relation to those individuals not developing pain (Group 1).**

| | Group 1 (n = 880) | Group 2 (n = 33) | p-value | RR (95%CI) |
|---|---|---|---|---|
| Age, mean ± SD | 45.2 ± 16.9 | 49.2 ± 14.3 | 0.172 | - |
| BMI (kg/m$^2$), mean ± SD | 25.2 ± 6.5 | 28.1 ± 9.0 | 0.072 | - |
| Female, n (%) | 503(57.2) | 25 (75.8) | 0.034 | 1.3 (1.1–1.6)[a] |
| Marital status, n (%) | | | 0.546 | - |
| Married | 482 (54.8) | 20 (60.6) | - | - |
| Divorced / Widow | 80 (9.1) | 4 (12.1) | - | 1.2 (0.5–2.9) |
| Single | 318 (36.1) | 9 (27.3) | - | 0.8 (0.5–1.4) |
| Graduated high-school and below, n (%) | 648 (73.6) | 27 (81.8) | 0.293 | 0.7 (0.3–1.4) |
| Pre-existing comorbidities, n (%) | 366 (41.6) | 21 (63.6) | 0.012 | 1.5 (1.2–2.0)[a] |
| Mental health problems, n (%) | 17 (1.9) | 1 (3.0) | 0.656 | 1.6 (0.3–11.4) |
| Severity of COVID-19 infection, n (%) | | | 0.013 | - |
| Mild | 813 (92.4) | 26 (78.8) | - | - |
| Moderate | 36 (4.1) | 3 (9.1) | - | 2.4 (0.8–7.5) |
| Severe | 31 (3.5) | 4 (12.1) | - | 3.6 (1.4–9.6)[a] |
| Hospital admission, n (%) | 558 (63.4) | 24 (72.7) | - | 1.2 (0.9–1.4) |
| Length of stay (days), mean ± SD | 5.4 ± 5.6 | 6.6 ± 4.6 | 0.019 | - |
| Vaccinated before infection, n (%) | 601 (68.3) | 24 (72.3) | 0.591 | 0.8 (0.5–1.5) |
| Fatigue, n (%) | 165 (18.8) | 12 (36.4) | 0.012 | 1.9 (1.2–3.1) |
| Anosmia, n (%) | 63 (7.2) | 3 (9.1) | 0.727 | 1.3 (0.4–3.8) |
| EQ-5D-5L, utility score (0–1) mean ± SD | 0.97 ± 0.11 | 0.87 ± 0.17 | 0.004 | - |
| EQ-5D-VAS score (0–100), mean ± SD | 89.5 ± 11.3 | 77.55 ± 15.50 | <0.001 | - |

Abbreviation: BMI, Body Mass Index; 95% CI, 95% Confidence Interval; EQ-5D-5L, EuroQol 5 Dimensions 5 Levels; VAS, Visual Analog Scale; RR, Relative Risk; SD, Standard Deviation; PCCP, Post-COVID Chronic Pain.

[a]Statistically significant relative risk.

### Factors associated with chronic pain with aggravated pain post-COVID pain

From 106 chronic pain participants who had been infected with COVID-19, 38 participants (35.9%, 95% CI 27.4–45.3) reported aggravated pain after the infection. The comparison between chronic pain with COVID-19 aggravated pain (Group 3) and chronic pain without COVID-19 aggravated pain (Group 4) is shown in Table 6. After adjusting relative risk, the significant risks of experiencing aggravated pain after COVID-19 were single status (RR 1.8, 95% CI 1.1–3.1, p = 0.021) and mental health problems (RR 1.8, 95% CI 1.1–2.9, p = 0.018). There is no significant association between severity of infection, admission status, length of stay or vaccination status and the aggravated pain post-COVID.

### Discussion

The study found that the overall incidence of de novo post-COVID chronic pain (PCCP) in a Thai sample was 3.2%, ranging from 2.8 to 8.5% with the highest incidence for the most severe COVID-19 cases according to the WHO clinical progression scale. Female sex and higher COVID-19 severity were factors associated with developing de novo PCCP. The incidence of chronic pain worsening due to COVID-19 was found to be 3.7% (95% CI 2.7–5.1) among

**Table 6. The Relative risk of experiencing exacerbated chronic pain following COVID-19 (Group 3) compared to experiencing chronic pain without exacerbation.**

| | Group 3 (n = 38) | Group 4 (n = 68) | p-value | RR (95%CI) |
|---|---|---|---|---|
| Age, mean ± SD | 51.52 ± 16.69 | 59.13 ± 16.44 | 0.025 | - |
| BMI (kg/m$^2$), mean ± SD | 24.91 ± 5.81 | 26.76 ± 5.43 | 0.105 | - |
| Female, n (%) | 27 (71.1) | 46 (67.6) | 0.717 | 1.1 (0.8–1.4) |
| Thai, n (%) | 35 (92.1) | 65 (95.6) | 0.664 | 1.0 (0.9–1.1) |
| Marital status, n (%) | | | 0.005 | - |
| Married | 21 (55.3) | 41 (60.3) | - | - |
| Divorced / Widow | 5 (13.2) | 21 (30.9) | - | 0.6 (0.2–1.3) |
| Single | 12 (31.6) | 6 (8.8) | - | 2.9 (1.2–6.8)[a] |
| Graduated high-school and below, n (%) | 27 (71.1) | 49 (72.1) | - | 1.0 (0.8–1.3) |
| Underlying disease, n (%) | 21 (55.3) | 50 (73.5) | 0.055 | 0.8 (0.6–1.0) |
| Mental health problems, n (%) | 8 (21.1) | 5 (7.4) | 0.061 | 2.9 (1.0–8.1)[a] |
| Severity of COVID-19 infection, n (%) | | | 0.117 | - |
| Mild | 26 (68.4) | 49 (72.1) | - | - |
| Moderate | 10 (26.3) | 9 (13.2) | - | 1.8 (0.8–4.0) |
| Severe | 2 (5.3) | 10 (14.7) | - | 0.4 (0.1–0.2) |
| Hospital admission, n (%) | 25 (65.8) | 54 (79.4) | 0.123 | 0.8 (0.6–1.1) |
| Duration from infection to interview (days), mean ± SD | 139.84 ± 23.93 | 138.02 ± 18.77 | 0.689 | - |
| Length of stay (days), mean ± SD | 6.7 ± 5.9 | 7.7 ± 5.8 | 0.376 | - |
| Vaccinated before infection, n (%) | 19 (50.0) | 35 (51.5) | 0.885 | 1.0 (0.7–1.4) |
| Fatigue, n (%) | 20 (52.6) | 23 (33.8) | 0.059 | 1.6 (1.0–2.4)[a] |
| Anosmia, n (%) | 4 (10.5) | 2 (2.9) | 0.184 | 3.6 (0.7–18.6) |
| EQ-5D-5L, utility score (0–1) mean ± SD | 0.78 ± 0.25 | 0.82 ± 0.21 | 0.460 | - |
| EQ-5D-VAS score (0–100), mean ± SD | 72.3 ± 18.24 | 76.32 ± 13.32 | 0.198 | - |

Abbreviation: BMI, Body Mass Index; 95%CI, 95% Confidence Interval; EQ, EuroQol 5 Dimensions 5 Levels; VAS, Visual Analog Scale; RR, Relative Risk; SD, Standard Deviation.

[a]Statistically significant relative risk.

individuals with a history of chronic pain. Within the chronic pain group, this incidence increased to 35.9% (95% CI 27.4–45.3). Factors associated with this aggravation included mental health issues and single-marital status.

## The incidence of de novo post-COVID chronic pain in population in Thailand

Post-COVID pain is now recognized as a common symptom after an acute SARS-CoV-2 infection [1,4,5,21]. However, the current study found a lower incidence of de novo PCCP (3.2%) in the Thai population as compared with the Brazilian (19.6%) [6], Iranian (15.3%) [22] or European (17%) [10] populations. The difference in incidence in our study can be explained by the following factors.

First, the population in most previous studies have focused selectively on previously hospitalized COVID-19 survivors with greater disease severity, whereas the current population included mostly participants with mild symptoms (89.7%), and only 64.9% were hospitalized. In fact, we found that the incidence of PCCP was higher according to COVID-19 severity; therefore, the lower overall incidence in our study could partly be explained by the lower proportion of severe cases.

The second explanation is possibly due to the different race, ethnicity, and genetic factors. A study from Japan observed that 10.4% of patients reported persistent pain during the pandemic, while only 6.3% of participants had pre-existing chronic pain [2]. Previous research suggests that race and ethnicity are essential considerations when investigating chronic pain [23]. There were studies reported that chronic pain prevalence in Japan, Pakistan, Thailand, and Myanmar was 17.5%, 15.8%, 19.9%, and 5.9%, respectively [24,25]. Moreover, chronic pain prevalence in Denmark, UK and USA was 27.8%, 35%-51% and 20%, respectively [26–28]. Zajacova et al. found that Asian Americans exhibited significantly lowest pain prevalence rates [29]. The underlying causes of disparities in chronic pain are complex, and differences in pain beliefs, cognitions, and behavior between ethnic groups might impact low incidences of chronic pain.

Genetic variations may influence pain sensitivity and perception, such as catechol-O-methyltransferase (COMT) and the μ-opioid receptor gene (OPRM1) [30]. However, there was no connection between genetic variation related to COVID-19 and the presence of long-lasting COVID-19 symptoms [31]. Epigenetics mechanism might explain racial disparities in chronic pain [32]. However, the correlation between race, ethnicity, and chronic post-COVID pain has not been investigated in detail.

## Risk factors for development of de novo post-COVID chronic pain in a population in Thailand

The current study found that female sex and greater COVID-19 severity were risk factors associated with the development of post-COVID chronic pain in Thai participants. Fernandez-de-las-Peñas et al. reported that female sex, history of previous musculoskeletal pain, myalgia and headache at hospitalization and days at the hospital were risk factors associated with post-COVID musculoskeletal pain in Spanish hospitalized COVID-19 survivors [33]. In fact, female sex seems to be a clear risk factor for overall post-COVID symptomatology [34]. Hormones may play a role in hyper-inflammatory status during the acute phase of SARS-CoV-2 infection, and the effect may remain after recovery [35]. Zeng et al. reported a relatively higher concentration of serum SARS-CoV-2 immunoglobulin G (IgG) antibodies in female than in male infected by SARS-COV-2 [36]. Nevertheless, the association between serological biomarkers and the development of post-COVID pain regarding sex is not clear [37].

Chronic pain conditions have also a bidirectional relationship with psychological problems, as one can lead to the other due to shared biological mechanisms [38]. Related to further stressors during acute SARS-CoV-2 infection, including social distancing, isolation, and quarantine, those with underlying mental health disorders are at higher risk for exacerbating their mental health problems and could promote chronic pain [39]. Mazza et al. evaluated the association between SARS-CoV-2 infection and psychiatric implications and found a considerable proportion of patients in the psychopathological range, including post-traumatic stress disorder (PTSD), depression, anxiety, and insomnia [40]. A mental health underlying condition is also a strong predictor for developing pain after COVID-19 illness and long COVID symptoms [21,41]. Bileviciute-Ljungar et al. investigated pain burden in sufferers of post-COVID disease and indicated that medical comorbidities might play a role in widespread pain [42].

We also found that severe COVID-19 illness was associated with a higher incidence of PCCP in a Thai population. Magdy et al. assessed risk factors associated with persistent neuropathic pain in COVID-19 survivors and reported that moderate and severe COVID-19 condition were significantly higher in those with post-COVID pain [43]. Nevertheless, Bai et al. reported no association between COVID-19 severity and long-COVID. Furthermore, there was no difference between orotracheal intubation/continuous positive airway pressure/non-

invasive mechanical ventilation versus no $O_2$ therapy [35]. Anaya et al. also reported that the incidence of post-COVID syndrome, including musculoskeletal pain, was unrelated to the severity of acute illness [44]. Soares et al. showed that new-onset fatigue was more common in COVID-19 survivors reporting post-COVID pain [6]. However, our study also explored these post-COVID symptoms and found that only de novo PCCP was associated with post-COVID fatigue but not with the presence of anosmia.

### Aggravated post-COVID pain symptoms in individuals with pre-COVID chronic pain

Another interesting finding in our study is that we also analyzed the presence of aggravated post-COVID pain in those participants with previous chronic pain conditions (Group 3). The incidence of aggravated post-COVID pain in a Thai population was 3.7% (95% CI 2.7–5.1) and 35.9% (95% CI 27.4–45.3) among the chronic pain participants. Additionally, this group reported the worst pain intensity, trend toward worst interferences, and lowest health-related quality of life compared to those developing de novo PCCP (Group 2). Chronic pain is a common condition that impacts millions of people worldwide, and our result showed that one-third of chronic pain participants may develop aggravated pain. The presence of aggravated pain after COVID-19 will significantly impact the population, society, healthcare services, and the economy. Patients suffering from chronic pain are considered vulnerable to the impact of long-COVID [45] not only due to high prevalence of aggravated pain after infection but also the lowest quality of life among all groups. Additionally, as patients with previous mental problems and being single have higher risk factors for the presence of aggravated post-COVID pain, early identification of individuals with these risk factors may lead to early detection and prevention of the impact of aggravated pain after COVID-19. Accordingly, strategies to prevent this consequence, including primary prevention (by promoting COVID-19 vaccination among chronic pain patients), secondary prevention (early detection of aggravated post-COVID pain in chronic pain patients), and tertiary prevention (rehabilitation strategies and prevention disability) should be implemented.

### Limitations and future research

Although this is the first study specifically investigating the development of post-COVID pain in a Thai population, our limitations include that this was a single-center cross-sectional study, and we collected data through a telephone interview with no face-to-face investigation. While a review supported telephone interviews for potentially reducing response bias compared to in-person interviews and enhancing participants' reporting accuracy [46], the study's context imposed limitations on physical examinations and the ability to pinpoint the exact pain location. Moreover, some symptoms, such as fatigue, should be assessed with a specific questionnaire or in-person examination to really diagnose post-COVID fatigue. Nevertheless, albeit these potential limitations, current results suggest that future research should explore the trajectory of symptoms, as it has been conducted in Spanish COVID-19 survivors [47]. Unlike the current study, which primarily reported pain symptoms and location, future research should extend its scope to draw conclusive pain diagnoses and incorporate comprehensive physical examinations of the neurological and musculoskeletal systems in the Thai cohort.

### Conclusions

The study revealed the overall incidence of de novo post-COVID chronic pain (PCCP) in Thailand, ranging from 2.8% to 8.5%, with the highest incidence for the most severe COVID-19 cases according to the WHO clinical progression scale. Musculoskeletal pain was the most

common diagnosis, followed by primary headache. Female sex and severe COVID-19 disease were factors associated with de novo post-COVID chronic pain. The overall incidence of previous chronic pain aggravated by COVID-19 was 35.9% (95% CI 27.4–45.3), and having mental problems and being single were risk factors for aggravated pain. The chronic pain patients with aggravated pain reported the highest pain intensity and trend toward worst pain interference and quality of life among patients who experienced pain after COVID-19.

## Supporting information

**S1 Checklist. STROBE statement—Checklist of items that should be included in reports of observational studies.**
(DOCX)

**S1 File.**
(DOCX)

**S1 Appendix. Raw data of included participants.**
(XLSX)

**S2 Appendix. Interview script and questionaire.**
(PDF)

**S3 Appendix. Vaccine types and pain interferences.**
(DOCX)

## Acknowledgments

The authors gratefully acknowledge the patients who generously agreed to participate in this study, Ms. Nattaya Bunwatsana for general research assistance, Ms. Julaporn Pooliam for statistical analysis, and Professor Andrew Rice for the valuable comments.

## Author Contributions

**Conceptualization:** Suratsawadee Wangnamthip, Nantthasorn Zinboonyahgoon, Pranee Rushatamukayanunt, Panuwat Promsin, Rujipas Sirijatuphat.

**Formal analysis:** Suratsawadee Wangnamthip, Nantthasorn Zinboonyahgoon.

**Investigation:** Patcha Papaisarn, Burapa Pajina, Thanawut Jitsinthunun, Panuwat Promsin, Rujipas Sirijatuphat.

**Methodology:** Suratsawadee Wangnamthip, Nantthasorn Zinboonyahgoon, Pranee Rushatamukayanunt, Panuwat Promsin, Rujipas Sirijatuphat.

**Writing – original draft:** Suratsawadee Wangnamthip, Nantthasorn Zinboonyahgoon, Patcha Papaisarn.

**Writing – review & editing:** Suratsawadee Wangnamthip, Nantthasorn Zinboonyahgoon, César Fernández-de-las-Peñas, Lars Arendt-Nielsen, Daniel Ciampi de Andrade.

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
