## [Decision Letter · Decision Letter 0]

8 Sep 2023

PONE-D-23-21908The incidence, characteristics, impact and risk factors of post-COVID chronic pain in Thailand: a single-center cross-sectional studyPLOS ONE

Dear Dr. Zinboonyahgoon,

Thank you for submitting your manuscript to PLOS ONE. After careful consideration, we feel that it has merit but does not fully meet PLOS ONE’s publication criteria as it currently stands. Therefore, we invite you to submit a revised version of the manuscript that addresses the points raised during the review process.

We look forward to receiving your revised manuscript.

Kind regards,

Amin Nakhostin-Ansari

Academic Editor

PLOS ONE

Journal Requirements:

Reviewers' comments:

Reviewer's Responses to Questions

**Comments to the Author**

1. Is the manuscript technically sound, and do the data support the conclusions?

Reviewer #1: Yes

Reviewer #2: Yes

2. Has the statistical analysis been performed appropriately and rigorously? 

Reviewer #1: Yes

Reviewer #2: Yes

3. Have the authors made all data underlying the findings in their manuscript fully available?

Reviewer #1: Yes

Reviewer #2: Yes

4. Is the manuscript presented in an intelligible fashion and written in standard English?

Reviewer #1: Yes

Reviewer #2: Yes

5. Review Comments to the Author

Reviewer #1: This comprehensive study examines the incidence of chronic pain related to SARS=CoV-2 infection. Data collected from a single centre cross-sectional study from Thailand reports the incidence, characteristics and risk factors for post-COVID-19 chronic pain (PCCP) for 1,019 patients with a confirmed diagnosis of SARS-CoV-2 infection.

The incidence of overall chronic pain was 3.2%, but ranged from 2.8%-8.5% for different severity of infection, with female sex and severity of infection identified as risk factors. Of particular interest is that 35.9% or 1/3 of those with a prior history of chronic pain had aggravation of chronic pain related to infection, with single status and mental health problems identified as risk factors.

The methodology is clearly stated and follow up was between 3-6 months (about 140 days) after infection. Data on demographics, premorbid health and vaccine status, and COVID related symptoms was collected via telephone interview. Of 1490 potential participants, data was obtained from 1019, with full explanation for drop outs. Severity of infection was identified and included those admitted as well as those managed at home.

The paper is well written, easy to follow and well referenced. Authors have provided an excellent discussion, made suggestions for particular findings and fully acknowledged limitations.

I have no additional comments or suggestions.

Reviewer #2: In this study the authors evaluated the incidence and impact of chronic pain in a Thai patient sample, several months after the COVID infection. Data was collected via telephone interviews and it allowed to determine the major risk factors for developing chronic pain after COVID, as well as determining the factors that seem to aggravate the condition in those who had it prior to the infection. The authors identified female sex and COVID-19 severity to be the major risk factors for developing post COVID chronic pain, while being single and having mental health problems were implied in previously existing chronic pain worsening.

Firstly, I would like to start by identifying some of the positive aspects I have found in the present work, those being:

• I was very pleased with the fact that the raw data was made easily accessible and possible to download, this shows integrity and professionalism of the authors.

• On the same line, it is positive that the authors followed the STROBE guidelines for reporting their work.

• The ethical aspects were duly described.

• The definition of pain used was given. Authors followed ICD-11 revision as guide to defining chronic pain.

• Authors also explained why particular post-covid symptoms were assessed.

• Recruitment and data collection diagram facilitates a rapid reading of the paper.

• The authors adjusted RR with linear models, limiting confounding effects.

• The authors acknowledge the importance of beliefs, cognitions and behaviours on what concerns chronic pain incidence and experience.

• The authors describe ways in which chronic pain aggravation due to COVID infection can be prevented, on several levels.

• All authors contributions are properly disclosed.

• References are very recent and it indicates de work is integrated in the current state of the art.

Despite the positive aspects, there are also some issues I have found in this work that deserve to be clarified/addressed. Below are all the commentaries, made by section and numerated, so that they are easy to identify and for the authors to address and/or respond:

»» Abstract

1. P. 3 – line 49: “This study aimed to study…”, redundant, might deserve rewording.

2. P. 3 – line 53: “(…) and phone interviews were done at between 3 to 6 months post-infection.(…), consider removing “at”.

3. P-3 – lines 58-59: “(…) The overall incidence of PCCP was 3.2% (95% CI 2.3-4.5), with of 2.8% (95% CI 2.0-4.1), 5.2% (…)”, there is a word missing before “of 2.8%” and it renders the phrase confusing.

4. P. 3 – lines 63-66: These 2 phrases are repetitive as they describe the exact same information already given above. Consider rewriting the conclusions paragraph.

5. As the title mentions the impact of chronic pain was evaluated, I think it should be mentioned in a succinct phrase in the abstract. The same is valid for quality of life evaluation.

»» Introduction

6. P. 4 – lines 70 to 74: the English is rather poor in this paragraph; please consider reviewing.

7. P. 4 – line 78: “(…) As such pain is a critical consequence (…)”, sounds as too much emphasis on this post-COVID symptom, authors might consider a more toned classification.

8. P. 5 – lines 100 to 102: the authors enumerate pain characterization, pain severity and pain impact on daily life as separated items but the last two are part of the characterization of pain. This presentation is somewhat confusing and persists throughout the paper, please review how this points are presented to the reader.

9. P. 5 – line 102: “(..) and 3) to identify those risk factors (…)”, consider substituting “those” with “the”.

»» Materials and methods

10. P. 6 – line 119 and 120: authors mention the application of “standardized conversations and questionnaires” but do not provide an example of either of them. This is a major issue to address as it renders the present study difficult to be replicated - please consider providing access to the interview and conversation guides used in the study.

11. P. 6 – lines 123 to 124: “ Participants were assessed through telephone interviews with standardized trained interviewers to reduce non-response bias (…)”: please elaborate on what are standardized trained interviewers. Authors may explain what the training consist-ed of, for example, so that it can be replicated.

12. P. 6 – line 128: “Calls were randomly chosen and conducted from (…)”; please be more clear on what was randomly chosen (was it the telephone numbers?) and how it was randomly chosen.

13. P. 6 and 7 – lines 133 to 136: it would be interesting to briefly explain why these were the selected items to collect data on (to render the study comparable to existing literature or other aspects. Authors may give references).

14. P. 7 – lines 148 to 149 and 152: “Symptoms experienced after the acute infection were self-reported by the participants at the time of assessment, including pain, fatigue and anos-mia.(…)” and “(…)Participants answered yes/no to the presence of these symptoms. (…)”; given that participants were questioned if they had a specific symptom with closed ques-tions (yes or no), I think describing data collection as “self-reported” is not adequate; please consider rewording.

15. P. 8 – line 163 and 164: the authors say “Participants in Groups 2-4 were also interviewed about their pain characteristics, location and type”; again, the location and type of pain are part of pain characterization, this is slightly confusing.

16. P. 8: lines 176 to 180: authors may want to explain why they chose to use two different quality of life questionnaires; they are clearly different but it would be interesting to know why they deemed it necessary to obtained two different assessments of the same topic.

17. P. 9 – line 184: consider substituting “(…) the last 24 hours during general activity, (…)” for “(…) the last 24 hours regarding general activity, (…). This is a minor issue.

18. P. 10 – lines 214 and 215: it is a positive aspect to explain how missing data was dealt with but I think this phrases deserve rewording, they give the appearance of two different things.

»» Results

19. P. 10: lines 221 and 222: “(…) The majority causes of dropout were unable to (…)”, please consider rewording, for example “(…) Major causes of dropout were inability to (…).

20. Figure 1: it is important to mention how many cases were removed from the final analysis because of missing data, it is not clear.

21. P. 11: Table 1: I would suggest the authors to separate demographic data from the clinical data of participants, that is, creating 2 separate (and smaller, easier to read) tables. Also, given the absence of relevance of type of vaccine to the study, this information could be provided in a supporting information file, rendering the table less noisy.

22. P. 13 – lines 240 to 242: authors may want to review wording, is a bit difficult to follow. Also, authors mention that “(…) the incidence of PCCP according to COVID-19 severity was 2.8% (95% CI 2.0-4.2) in those with mild disease, 5.2% (95% CI 1.2-14.2) in those with moderate and 8.5% (95% CI 3.4-19.9) in those who had severe illness (…).” The incidence values referred here (which are repeated throughout the paper) aren’t present in any table, which is the main tool for resuming information in this study. I strongly suggest that incidence of PCCP according to COVID severity is included in Table 1, under the line “2. De novo Post-COVID Chronic Pain (PCCP)”.

23. P. 14 – Table 2: minor suggestion, authors could give a more succinct title to this table and include group description inside the table. Also, again the type of vaccine given is present and its relevance to the study is debatable (it was not mentioned in the rest of the study), could be provided in a supporting information file.

24. P. 17 – lines 272 to 274: concerning pain localization, this last paragraph is just enough and there is no need for a figure to accompany it. This is merely an aesthetic option but I think that, if the authors choose to use an image they could change the one used here for it is not that easy to interpret and is quite flashy. I suggest the authors consider using something in the lines of the imaging used by an Author cited in the study: Soares et. al, 2021.

25. P. 17 to 19 – Table 3: the pain interference (note that the title says “interference” and in the table it says “interferences”, should be harmonized) is rather extensively depicted in the table, but the description in lines 288 and 289 (P. 19) is sufficient to describe this results. The details could be provided in a supporting information file.

26. P. 19 – line 289: “(…) did not reach statistical significance (Table 2).”, this information is presented in Table 3, not Table 2, please correct.

27. P. 19 – lines 291 and 292: consider if it really is necessary to include the information in parenthesis, it renders the subtitle too long.

28. P. 20 – Table 4: consider the suggestion made for Table 2 (see commentary 23).

29. P. 23 – Table 5: beware of the English of this title, it is quite repetitive. I suggest the following alternative “Relative risk of suffering from aggravated chronic pain after COVID-19 in relation to chronic pain without aggravation”, including the group description inside the table, as suggested previously in other commentaries.

»» Discussion

30. P. 25 – lines 341 to 343: please review the wording and the phrase construction of the third paragraph, it is not clear.

31. P. 25 – lines 343 to 345: the last paragraph simply describes results once more, please discuss if there is and what is the meaning of this findings.

32. P. 25 – lines 340 to 342: the authors refer overall incidence of 3.7% (regarding the total of participants) and 35.9% (regarding chronic pain participants) but the phrase construction makes it prone to confusion when reading. Consider rewording this information.

33. P. 25 – lines 349 and 350: I suggest authors to indicate more references that support the statement that post-COVID pain is recognized as a common symptom.

34. P. 27 – lines 363 to 365: the references cited are 10 or more years old (reference 20 and 21), authors are advised to include more recent studies (there is no need to remove the used ones).

35. P. 26 and 27 – lines 370 to 381: although genetic differences are important regarding chronic pain, the authors go into too much detail discussing this topic, which acquires an importance that seems to be much higher than the previous discussed factors. In my opinion, the genetic particularities should be discussed in studies in the field of genetics and/or basic investigations, therefore, this part of the discussion should be much more succinct and direct, one or two phrases at most. I suggest the authors reformulate this part.

36. P. 27 – lines 382 to 386: as the authors comment themselves, the majority of the studies compared were also conducted via telephone interviews, therefore, this factor does not explain the difference in incidence obtained in this study. The paragraph is contradictory, please review.

37. P. 28 – lines 399 and 400: the references cited state that the is no association between serological biomarkers and development of post-COVID pain, they do not say that there is an unclear association. Also, neither of the references analysed serological biomarkers regarding female or male sex, which makes them irrelevant in this discussion. I must also state that both references are by the same author, whom is also author of the submitted study, when there are other references available in the literature (as possible examples: doi.org/10.3389/fmed.2023.1085988; doi: 10.3390/v15081724).

38. P. 28 – lines 402 to 405 and lines 408 and 409: include references.

39. P. 28 – lines 416 to 418: please explain why so much highlight was given to the need of supplementary oxygen in particular? There are several other aspects related to COVID infection that indicate its severity.

40. P. 29 – lines 432 to 435: this phrase is confusing; using the one-third in this way seems to establish a comparison that has no place. Please review the wording here.

41. P. 29 – lines 435 to 438: it would be interesting if the authors went further discussing the greater vulnerability of chronic pain patients, as it is central in this study, and also give more references in this topic. Pay attention to the words used in the two phrases; the connector “in fact (…)” implies that having previous mental problems and being single is somehow related to having chronic pain, which constitutes an overreach in my opinion.

42. Although the authors used two different questionnaires for assessing quality of life, differences between results are not addressed. It would be interesting to have some discussion regarding this topic to help understand the reasoning behind the decision to apply to instruments.

43. As general rule the type of vaccination is present in almost all Tables but is never referred in Discussion or Results.

»» Limitations

44. P. 30 – line 447: explain further the telephone interviews limitations compared to face-to-face interventions in the context of this study.

45. P. 30 – lines 447 to 449: explain what is the effect of this limitation – do the authors expect results to be over or underestimated?

46. P. 30 – lines 449 to 452: please explain with more clarity how/why the results of the study suggest that the trajectory of symptoms should be explored.

47. Authors could mention the fact that a great portion of patients who have COVID did not use health services, therefore, are left out of this type of study, affecting results.

»» Conclusion

48. P. 30 – lines 461 to 463: this phrase reinforces the importance of addressing patient vulnerability previous to infection. The authors could further discuss this topic and its variables, as suggested for chronic pain patients in commentary number 41.

49. Please review the language cohesiveness of the Conclusion.

»» References

50. P. 35 – lines 559 to 566: references 26 and 27 are the same paper, please confirm.

51. One of the authors of this study is cited 10 times in a total of 42 references. Take into consideration the commentaries made above and please review the references used.

As a general summary, the key points to the authors are:

- Revise the language used and the length of many of the Table titles and subtitles.

- To try to summarize the results in smaller, easier to interpret tables; there is no need to give so much detail in the paper when some of the information can be available online and/or in a supporting information file.

- Type of vaccine is included in every table but is never mentioned in Results or Discussion; it either matters or is irrelevant information that could be made available through a supporting information document.

- To further discuss certain topics (e.g. patient vulnerability) in detriment of others that do not seem so relevant in this context (e.g. genetics).

- Review the references thoroughly and pay special attention to self-citations and reference repetition.

All in all, although the study itself is not new in terms of investigation query or methodology, it does provide new information on a specific topic and population which there is still not much available (i.e. Asian patients). It is a work in line with others already published but with novel information, therefore, I consider this to be valuable and relevant knowledge in the field.

I hope the authors take the commentaries above as constructive observations that aim at helping them with the revisions that need to be made for the paper to be eligible to publication.

6. PLOS authors have the option to publish the peer review history of their article (what does this mean?). If published, this will include your full peer review and any attached files.

Reviewer #1: No

Reviewer #2: No

---

## [Author Response · Author response to Decision Letter 0]

21 Nov 2023

Dear Reviewer #2

We would like to thank you for carefully reviewing our manuscript, the positive comments about the contribution of the findings, and the suggestions for improving the paper further. Below we present a point-by-point response to each suggestion, including the changes made in the paper to address each one. All changes made in the text are indicated in yellow highlighted.

»» Abstract

1. P. 3 – line 49: “This study aimed to study…”, redundant, might deserve rewording.

Author response: Thank you for your comment. We have reworded it to not be redundant on page 3, lines 49-50 of the abstract.

2. P. 3 – line 53: “(…) and phone interviews were done at between 3 to 6 months post-infection.(…), consider removing “at”.

Author response: Thank you for your suggestion. We have removed "at” from page 3, line 53. 

3. P-3 – lines 58-59: “(…) The overall incidence of PCCP was 3.2% (95% CI 2.3-4.5), with of 2.8% (95% CI 2.0-4.1), 5.2% (…)”, there is a word missing before “of 2.8%” and it renders the phrase confusing.

Author response: Thank you for your suggestions. We have revised on page 3, lines 58-60.

4. P. 3 – lines 63-66: These 2 phrases are repetitive as they describe the exact same information already given above. Consider rewriting the conclusions paragraph.

Author response: Thank you for your suggestion. We have revised lines 65-68 of the abstract.

5. As the title mentions the impact of chronic pain was evaluated, I think it should be mentioned in a succinct phrase in the abstract. The same is valid for quality of life evaluation.

Author response: Thank you for your suggestion. We have revised page 3, lines 63-65 of the abstract, to mention participants' quality of life and pain interference.

»» Introduction

6. P. 4 – lines 70 to 74: the English is rather poor in this paragraph; please consider reviewing.

Author response: Thank you for your suggestion. We have revised the paragraph on page 4, lines 70-76.

7. P. 4 – line 78: “(…) As such pain is a critical consequence (…)”, sounds as too much emphasis on this post-COVID symptom, authors might consider a more toned classification.

Author response: Thank you for your suggestion. We have reworded on page 4, lines 80-81.

8. P. 5 – lines 100 to 102: the authors enumerate pain characterization, pain severity and pain impact on daily life as separated items but the last two are part of the characterization of pain. This presentation is somewhat confusing and persists throughout the paper, please review how this points are presented to the reader.

Author response: Thank you for your suggestion. We have revised the sentence on page 5, lines 102-103.

9. P. 5 – line 102: “(..) and 3) to identify those risk factors (…)”, consider substituting “those” with “the”.

Author response: Thank you for your suggestion. We have changed “those” to “the” on page 5, line 103.

» Materials and methods

10. P. 6 – line 119 and 120: authors mention the application of “standardized conversations and questionnaires” but do not provide an example of either of them. This is a major issue to address as it renders the present study difficult to be replicated - please consider providing access to the interview and conversation guides used in the study.

Author response: Thank you for your suggestion. We have added the interview and conversation guides as supplementary documents. However, all documents are in Thai language. (Page 6, lines120-121)

11. P. 6 – lines 123 to 124: “Participants were assessed through telephone interviews with standardized trained interviewers to reduce non-response bias (…)”: please elaborate on what are standardized trained interviewers. Authors may explain what the training consisted of, for example, so that it can be replicated.

Author response: Thank you for your suggestion. We have added the details of the training telephone interview and revised on page 6, lines 124-131)

12. P. 6 – line 128: “Calls were randomly chosen and conducted from (…)”; please be more clear on what was randomly chosen (was it the telephone numbers?) and how it was randomly chosen.

Author response: Thank you for your suggestion. We have added a method of random on page 6, lines 132-133. 

13. P. 6 and 7 – lines 133 to 136: it would be interesting to briefly explain why these were the selected items to collect data on (to render the study comparable to existing literature or other aspects. Authors may give references).

Author response: Thank you for your suggestion. We have revised and added a reference on page 7, lines 139-143. 

14. P. 7 – lines 148 to 149 and 152: “Symptoms experienced after the acute infection were self-reported by the participants at the time of assessment, including pain, fatigue and anos-mia.(…)” and “(…)Participants answered yes/no to the presence of these symptoms. (…)”; given that participants were questioned if they had a specific symptom with closed ques-tions (yes or no), I think describing data collection as “self-reported” is not adequate; please consider rewording.

Author response: Thank you for your suggestion. We have reworded from “self-report” to “interviewed” on page 7, line 154. 

15. P. 8 – line 163 and 164: the authors say “Participants in Groups 2-4 were also interviewed about their pain characteristics, location and type”; again, the location and type of pain are part of pain characterization, this is slightly confusing.

Author response: Thank you for your suggestion. We have revised the text on page 8, lines 169-170.

16. P. 8: lines 176 to 180: authors may want to explain why they chose to use two different quality of life questionnaires; they are clearly different but it would be interesting to know why they deemed it necessary to obtained two different assessments of the same topic.

Author response: Thank you for your comments. We utilized the EQ-5D-5L questionnaire to assess participants' quality of life, including EQ-utility and EQ-VAS components. The EQ-VAS, a scale ranging from 0 to 100, measures patients' self-assessed overall health on the day of questionnaire completion. According to the official EQ-5D-5L manual (page 21), EQ-VAS represents the patient's perspective, in contrast to EQ-5D utility, which reflects the value assigned to an EQ-5D profile based on a set of weights that capture general preferences regarding the profile's quality. Our analysis revealed no statistically significant disparities between the two outcomes. Nonetheless, it was evident that EQ-VAS scores were impacted by factors such as perceived locus of control, age, educational background, ethnic origin, and smoking habits, as indicated in the references."

https://euroqol.org/publications/user-guides/

https://euroqol.org/eq-5d-instruments/eq-5d-5l-about/faqs/

Whynes DK; TOMBOLA Group. Correspondence between EQ-5D health state classifications and EQ VAS scores. Health Qual Life Outcomes. 2008 Nov 7;6:94. doi: 10.1186/1477-7525-6-94. PMID: 18992139; PMCID: PMC2588564.

17. P. 9 – line 184: consider substituting “(…) the last 24 hours during general activity, (…)” for “(…) the last 24 hours regarding general activity, (…). This is a minor issue.

Author response: Thank you for your suggestion. We have replaced “regarding” with “during” on page 9, line 190.

18. P. 10 – lines 214 and 215: it is a positive aspect to explain how missing data was dealt with but I think this phrases deserve rewording, they give the appearance of two different things.

Author response: Thank you for your suggestion. We revised the sentence on page 10, lines 219-220.

»» Results

19. P. 10: lines 221 and 222: “(…) The majority causes of dropout were unable to (…)”, please consider rewording, for example “(…) Major causes of dropout were inability to (…).

Author response: Thank you for your suggestion. We have reworded the sentence on page 11, lines 227-228

20. Figure 1: it is important to mention how many cases were removed from the final analysis because of missing data, it is not clear.

Author response: Thank you for mentioning the important point. We have revised on page 9, lines 226-228, to clarify how many participants were dropped out and how many were included in the final analysis.

21. P. 11: Table 1: I would suggest the authors to separate demographic data from the clinical data of participants, that is, creating 2 separate (and smaller, easier to read) tables. Also, given the absence of relevance of type of vaccine to the study, this information could be provided in a supporting information file, rendering the table less noisy.

Author response: Thank you for your suggestion. We have separated the table into demographic data (table 1) and clinical data (table 2).

22. P. 13 – lines 240 to 242: authors may want to review wording, is a bit difficult to follow. Also, authors mention that “(…) the incidence of PCCP according to COVID-19 severity was 2.8% (95% CI 2.0-4.2) in those with mild disease, 5.2% (95% CI 1.2-14.2) in those with moderate and 8.5% (95% CI 3.4-19.9) in those who had severe illness (…).” The incidence values referred here (which are repeated throughout the paper) aren’t present in any table, which is the main tool for resuming information in this study. I strongly suggest that incidence of PCCP according to COVID severity is included in Table 1, under the line “2. De novo Post-COVID Chronic Pain (PCCP)”.

Author response: Thank you for your suggestion. We have added the incidence of PCCP according to COVID-19 severity to Table 2 under the line PCCP incidence.

23. P. 14 – Table 2: minor suggestion, authors could give a more succinct title to this table and include group description inside the table. Also, again the type of vaccine given is present and its relevance to the study is debatable (it was not mentioned in the rest of the study), could be provided in a supporting information file.

Author response: Thank you for your suggestion. We removed the type of vaccines from the table and added them to the S3 Appendix.

24. P. 17 – lines 272 to 274: concerning pain localization, this last paragraph is just enough and there is no need for a figure to accompany it. This is merely an aesthetic option but I think that, if the authors choose to use an image they could change the one used here for it is not that easy to interpret and is quite flashy. I suggest the authors consider using something in the lines of the imaging used by an Author cited in the study: Soares et. al, 2021.

Author response: Thank you for your suggestion. We have decided to remove Figure 2 and instead provide a textual description.

25. P. 17 to 19 – Table 3: the pain interference (note that the title says “interference” and in the table it says “interferences”, should be harmonized) is rather extensively depicted in the table, but the description in lines 288 and 289 (P. 19) is sufficient to describe this results. The details could be provided in a supporting information file.

Author response: Thank you for your suggestion. We have removed details of pain interference from Table 4 (previous Table 3) and provided them in the S3 Appendix. 

26. P. 19 – line 289: “(…) did not reach statistical significance (Table 2).”, this information is presented in Table 3, not Table 2, please correct.

Author response: Thank you for your comment. We have corrected “Table 2” to Table 4 (previous table 3) on page 19, line 293.

27. P. 19 – lines 291 and 292: consider if it really is necessary to include the information in parenthesis, it renders the subtitle too long.

Author response: Thank you for your suggestion. We have removed the information in parentheses (page 19, line 295).

28. P. 20 – Table 4: consider the suggestion made for Table 2 (see commentary 23).

Author response: Thank you for your suggestion. We removed the type of vaccines from the table and added them to the S3 Appendix.

29. P. 23 – Table 5: beware of the English of this title, it is quite repetitive. I suggest the following alternative “Relative risk of suffering from aggravated chronic pain after COVID-19 in relation to chronic pain without aggravation”, including the group description inside the table, as suggested previously in other commentaries.

Author response: Thank you for your suggestion. We have revised the Table 6 legend (page 23, lines 323-324)

»» Discussion

30. P. 25 – lines 341 to 343: please review the wording and the phrase construction of the third paragraph, it is not clear.

Author response: Thank you for your suggestion. We have revised this phrase on page 25, lines 333-337.

31. P. 25 – lines 343 to 345: the last paragraph simply describes results once more, please discuss if there is and what is the meaning of this findings.

Author response: Thank you for your comment. We have removed the last paragraph on page 25, lines 337

32. P. 25 – lines 340 to 342: the authors refer overall incidence of 3.7% (regarding the total of participants) and 35.9% (regarding chronic pain participants) but the phrase construction makes it prone to confusion when reading. Consider rewording this information.

Author response: Thank you for your suggestion. We have revised this phrase on page 25, lines 333-337.

33. P. 25 – lines 349 and 350: I suggest authors to indicate more references that support the statement that post-COVID pain is recognized as a common symptom.

Author response: Thank you for your suggestion. We have added two more references to support the statement on page 25, line 342.

34. P. 27 – lines 363 to 365: the references cited are 10 or more years old (reference 20 and 21), authors are advised to include more recent studies (there is no need to remove the used ones).

Author response: Thank you for your valuable suggestion. We have updated the more recent references (page 26, lines 355-358). However, the prevalence of chronic pain in ASEAN is still limited.

35. P. 26 and 27 – lines 370 to 381: although genetic differences are important regarding chronic pain, the authors go into too much detail discussing this topic, which acquires an importance that seems to be much higher than the previous discussed factors. In my opinion, the genetic particularities should be discussed in studies in the field of genetics and/or basic investigations, therefore, this part of the discussion should be much more succinct and direct, one or two phrases at most. I suggest the authors reformulate this part.

Author response: Thank you for your suggestion. We reformulated and revised the paragraph on page 26, lines 362-367.

36. P. 27 – lines 382 to 386: as the authors comment themselves, the majority of the studies compared were also conducted via telephone interviews, therefore, this factor does not explain the difference in incidence obtained in this study. The paragraph is contradictory, please review.

Author response: Thank you for your valuable comment. We have removed this paragraph.

37. P. 28 – lines 399 and 400: the references cited state that the is no association between serological biomarkers and development of post-COVID pain, they do not say that there is an unclear association. Also, neither of the references analysed serological biomarkers regarding female or male sex, which makes them irrelevant in this discussion. I must also state that both references are by the same author, whom is also author of the submitted study, when there are other references available in the literature (as possible examples: doi.org/10.3389/fmed.2023.1085988; doi: 10.3390/v15081724).

Author response: Thank you for your suggestion. We have revised the sentence and added a reference you suggested on page 27, lines 380-381.

38. P. 28 – lines 402 to 405 and lines 408 and 409: include references.

Author response: Thank you for your suggestion. We added a reference you suggested on page 27, lines 383-386, and 389-390.

39. P. 28 – lines 416 to 418: please explain why so much highlight was given to the need of supplementary oxygen in particular? There are several other aspects related to COVID infection that indicate its severity.

Author response: Thank you for your valuable comment. According to the WHO Clinical Progression Scale, the severity of COVID-19 infection is graded with oxygen supplementation. “Mild” was described as having mild symptoms and not requiring oxygen supplementation. “Moderate” was described as moderate symptoms which require an oxygen cannula to maintain SpO2 > 94%. “Severe” was defined as severe symptoms and need to use of high flow nasal cannula, non-invasive ventilator or mechanical ventilator to support the symptoms or required vasopressor or inotropic drugs.

40. P. 29 – lines 432 to 435: this phrase is confusing; using the one-third in this way seems to establish a comparison that has no place. Please review the wording here.

Author response: Thank you for your suggestion. We have revised the sentence you suggested on page 28, lines 413-417.

41. P. 29 – lines 435 to 438: it would be interesting if the authors went further discussing the greater vulnerability of chronic pain patients, as it is central in this study, and also give more references in this topic. Pay attention to the words used in the two phrases; the connector “in fact (…)” implies that having previous mental problems and being single is somehow related to having chronic pain, which constitutes an overreach in my opinion.

Author response: Thank you for your suggestion. We have revised the sentence you suggested on page 28-29, lines 418-422.

42. Although the authors used two different questionnaires for assessing quality of life, differences between results are not addressed. It would be interesting to have some discussion regarding this topic to help understand the reasoning behind the decision to apply to instruments.

Author response: Thank you for your comments. We utilized the EQ-5D-5L questionnaire to assess participants' quality of life, including both EQ-utility and EQ-VAS components. The EQ-VAS, a scale ranging from 0 to 100, measures patients' self-assessed overall health on the day of questionnaire completion. According to the official EQ-5D-5L manual (page 21), EQ-VAS represents the patient's perspective, in contrast to EQ-5D utility, which reflects the value assigned to an EQ-5D profile based on a set of weights that capture general preferences regarding the profile's quality. Our analysis revealed no statistically significant disparities between the two outcomes. Nonetheless, it was evident that EQ-VAS scores were impacted by factors such as perceived locus of control, age, educational background, ethnic origin, and smoking habits, as indicated in the references."

https://euroqol.org/publications/user-guides/

https://euroqol.org/eq-5d-instruments/eq-5d-5l-about/faqs/

Whynes DK; TOMBOLA Group. Correspondence between EQ-5D health state classifications and EQ VAS scores. Health Qual Life Outcomes. 2008 Nov 7;6:94. doi: 10.1186/1477-7525-6-94. PMID: 18992139; PMCID: PMC2588564.

43. As general rule the type of vaccination is present in almost all Tables but is never referred in Discussion or Results.

Author response: Thank you for your suggestion. We have removed the vaccine types from the tables and added them to the S3 Appendix.

»» Limitations

44. P. 30 – line 447: explain further the telephone interview limitations compared to face-to-face interventions in the context of this study.

Author response: Thank you for your suggestion. We added the limitation phone interview in this study context on page 29, lines 432-433.

45. P. 30 – lines 447 to 449: explain what is the effect of this limitation – do the authors expect results to be over or underestimated?

Author response: Thank you for your suggestion. We added the limitation phone interview in this study context on page 29, lines 430-433.

46. P. 30 – lines 449 to 452: please explain with more clarity how/why the results of the study suggest that the trajectory of symptoms should be explored.

Author response: Thank you for your suggestion. We revised to clarify why pain trajectory should be explored on page 29, lines 438-440.

47. Authors could mention the fact that a great portion of patients who have COVID did not use health services, therefore, are left out of this type of study, affecting results.

Author response: Thank you for your valuable feedback. From August to September 2021, every COVID-19 patient, regardless of severity, was comprehensively registered within the government system, including our hospital. Consequently, our study's results encompassed patients across all levels of severity.". 

»» Conclusion

48. P. 30 – lines 461 to 463: this phrase reinforces the importance of addressing patient vulnerability previous to infection. The authors could further discuss this topic and its variables, as suggested for chronic pain patients in commentary number 41.

Author response: Thank you for your suggestion. We have revised the sentence you suggested on page 30, lines 447-451.

49. Please review the language cohesiveness of the Conclusion.

Author response: Thank you for your comment. We have revised the conclusion paragraph on pages 30, lines 443-451

»» References

50. P. 35 – lines 559 to 566: references 26 and 27 are the same paper. Please confirm.

Author response: Thank you for your comment. We have removed both of them as we revised the text according to the genetic difference on page 26, lines 362-367.

51. One of the authors of this study is cited 10 times in a total of 42 references. Take into consideration the commentaries made above and please review the references used.

Author response: Thank you for your important comment. As our co-authors are the world authority, with many publications on post-COVID pain, it is almost impossible not to mention their works in the discussion part. However, we tried our best to reduce the unnecessary citation.

---

## [Decision Letter · Decision Letter 1]

18 Dec 2023

The incidence, characteristics, impact and risk factors of post-COVID chronic pain in Thailand: a single-center cross-sectional study

PONE-D-23-21908R1

Dear Dr. Zinboonyahgoon,

We’re pleased to inform you that your manuscript has been judged scientifically suitable for publication and will be formally accepted for publication once it meets all outstanding technical requirements.

Kind regards,

Amin Nakhostin-Ansari

Academic Editor

PLOS ONE

Additional Editor Comments (optional):

Reviewers' comments:

Reviewer's Responses to Questions

**Comments to the Author**

1. If the authors have adequately addressed your comments raised in a previous round of review and you feel that this manuscript is now acceptable for publication, you may indicate that here to bypass the “Comments to the Author” section, enter your conflict of interest statement in the “Confidential to Editor” section, and submit your "Accept" recommendation.

Reviewer #1: All comments have been addressed

Reviewer #2: All comments have been addressed

2. Is the manuscript technically sound, and do the data support the conclusions?

Reviewer #1: Yes

Reviewer #2: Yes

3. Has the statistical analysis been performed appropriately and rigorously? 

Reviewer #1: Yes

Reviewer #2: Yes

4. Have the authors made all data underlying the findings in their manuscript fully available?

Reviewer #1: Yes

Reviewer #2: Yes

5. Is the manuscript presented in an intelligible fashion and written in standard English?

Reviewer #1: Yes

Reviewer #2: Yes

6. Review Comments to the Author

Reviewer #1: Two tiny comments: Line 164 and line 200 corrections are now less clear. I suggest to revert to the original

Reviewer #2: All my comments were adressed by the authors with satisfying alterarions/answers.

Regarding the 23.8% of self-citation, the authors gave a reasonable explanation, I have no more objections on this matter.

I would like to thank you for your work and for your answers.

7. PLOS authors have the option to publish the peer review history of their article (what does this mean?). If published, this will include your full peer review and any attached files.

Reviewer #1: No

Reviewer #2: No

---

## [Editor Report · Acceptance letter]

4 Jan 2024

PONE-D-23-21908R1 

PLOS ONE

Dear Dr. Zinboonyahgoon, 

I'm pleased to inform you that your manuscript has been deemed suitable for publication in PLOS ONE. Congratulations! Your manuscript is now being handed over to our production team.

Kind regards, 

on behalf of

Dr. Amin Nakhostin-Ansari 

Academic Editor

PLOS ONE